# MicroRNA in Ovarian Cancer: Biology, Pathogenesis, and Therapeutic Opportunities

**DOI:** 10.3390/ijerph16091510

**Published:** 2019-04-29

**Authors:** San-Nung Chen, Renin Chang, Li-Te Lin, Chyi-Uei Chern, Hsiao-Wen Tsai, Zhi-Hong Wen, Yi-Han Li, Chia-Jung Li, Kuan-Hao Tsui

**Affiliations:** 1Department of Obstetrics and Gynecology, Kaohsiung Veterans General Hospital, Kaohsiung 813, Taiwan; snchen@vghks.gov.tw (S.-N.C); litelin1982@gmail.com (L.-T.L.); cuchern@vghks.gov.tw (C.-U.C.); drtsai0627@gmail.com (H.-W.T.); nigel6761@gmail.com (C.-J.L); 2Department of Emergency Medicine, Kaohsiung Veterans General Hospital, Kaohsiung 813, Taiwan; rhapsody1881@gmail.com; 3Department of Recreation Sports Management, Tajen University, Pingtung 907, Taiwan; 4Institute of Biotechnology and Chemical Engineering, I-Shou University, Kaohsiung 840, Taiwan; 5Department of Obstetrics and Gynecology, National Yang-Ming University School of Medicine, Taipei 112, Taiwan; 6Department of Biological Science, National Sun Yat-sen University, Kaohsiung 804, Taiwan; 7Department of Marine Biotechnology and Resources, National Sun Yat-sen University, Kaohsiung 804, Taiwan; wzh@mail.nsysu.edu.tw; 8Marine Biomedical Laboratory and Center for Translational Biopharmaceuticals, Department of Marine Biotechnology and Resources, National Sun Yat-sen University, Kaohsiung 813, Taiwan; 9Department of Pathology & Immunology, Baylor College of Medicine, Houston, TX 77030, USA; jack3395lee@gmail.com; 10Department of Pharmacy and Master Program, College of Pharmacy and Health Care, Tajen University, Pingtung 907, Taiwan

**Keywords:** microRNA, ovarian cancer, diagnosis, prognosis

## Abstract

Ovarian cancer comprises one of the three major malignant tumor types in the female reproductive system. The mortality rate of this cancer is the highest among all gynecological tumors, with ovarian cancer metastasis constituting an important cause of death. Therefore, markers for disease prediction and prognosis are highly desirable for early diagnosis as well as for helping optimize and personalize treatment. Recently, microRNAs (miRNAs), which consist of short-sequence RNAs that do not encode a protein, have emerged as new biomarkers in the clinical diagnosis and treatment of ovarian cancer. By pairing with bases specific to the target messenger RNA (mRNA), miRNAs cause degradation of the target mRNA or inhibit its translation, thereby regulating various cellular processes including cell proliferation and adhesion. Increasing numbers of studies have shown that miRNA expression abnormality plays an important role in the development of ovarian cancer. In this review, we discuss the mechanisms of miRNA action, current research regarding their role in the suppression or promotion of ovarian cancer, and their use as markers for diagnosis of prognosis or as therapeutic targets for this disease. Finally, we present future perspectives regarding the clinical management of ovarian cancer and the role for miRNAs therein.

## 1. Introduction

Ovarian cancer constitutes a malignant tumor that seriously threatens women’s health worldwide. Though its incidence is lower than that of cervical and endometrial cancers, its rate of mortality is highest among all female gynecological tumors [1]. Notably, ovarian cancer exhibits no specific clinical manifestations in the early stage, resulting in the majority of patients presenting with metastasis and invading cancer cells at the time of diagnosis. Recent reports indicate that the five-year survival rate of ovarian cancer is 42.9%; nevertheless, over 80.0% of patients with advanced ovarian cancer will relapse, with the associated prognosis being extremely poor [2,3]. At present, the clinical treatment of ovarian cancer is based on tumor cytoreductive surgery, for which, with the assistance of six to eight courses of paclitaxel and platinum-based combination chemotherapy, the complete remission rate can reach 70–80% [4]. However, high recurrence rates and high drug resistance rates after relapse are among the most important causes of the high mortality of this disease. Therefore, the metastasis, diagnosis, and treatment of ovarian cancer represent urgent problems that need to be solved [5,6]. To this end, the analysis of the expression of microRNAs (miRNAs), which consist of short-sequence RNAs that do not encode a protein, represents a strategy not only to explore the correlation between miRNA and ovarian cancer but also provide new targets for the diagnosis, treatment, and prognosis of ovarian cancer [5,7]. 

## 2. miRNA and Ovarian Cancer

### 2.1. Biological Characteristics of miRNA

miRNAs, which are widely found in eukaryotes, comprise a class of non-coding single-stranded small-molecule RNAs of approximately 19–22 nucleotides in length encoded by endogenous genes [8]. The first confirmed microRNA was discovered during the study of *Caenorhabditis elegans*. In 1993, Lee et al. discovered lin-4 when studying defects in *C. elegans* development, which was partially complementary to the 3’ non-coding regions (3’UTRs) of the target messenger RNA (mRNA). miRNA binding to a target mRNA can lead to its cleavage or destabilization. In addition, miRNAs can induce translation inhibition in a protein-specific manner, which inhibits protein synthesis and regulates the development of nematodes [9]. Subsequently, human-related miRNAs were identified. More recently, with the deepening of miRNA research, it has been found that miRNAs are involved in the post-transcriptional regulation of the expression of over 60% of human genes, thus impacting cell development, differentiation, metabolism, aging, inflammation, and immune response. miRNAs also have a direct relationship with tumor development and progression [10,11,12,13,14].

#### 2.1.1. Molecular Structure Characteristics of miRNA

The precursor miRNA (pre-miRNA) often forms an intramolecular stem-loop structure. The mature miRNA itself does not contain an open reading frame. The 5’ end carries a phosphate group, whereas the 3’ end comprises a hydroxyl group. The miRNA can specifically recognize the mRNA of one or several target genes. The untranslated regions (UTRs) at the 3’ end are complementary to each other, with the degree of complementary pairing determining the functional characteristics toward the mRNA. Generally, once fully complementary paired, the mRNA of the target gene is degraded; alternatively, if pairing is partially complementary, the translation of the target gene mRNA is inhibited [15].

### 2.1.2. miRNA Synthesis:

The synthesis and processing of miRNAs are illustrated in Figure 1. The pri-miRNA is processed by a microprocessor to form a 60–70 bp hairpin-like small RNA (i.e., pre-miRNA precursor) in the nucleus. The microprocessor consists of a ribonuclease Drosha and a microprocessor complex subunit 8 (DGCR8) [16]. Like Dicer, Drosha belongs to the RNA polymerase III family and is a class of endonucleases that act on double-stranded RNA (dsRNA). Drosha is a processing center consisting of two RNA polymerase III domains and a dsRNA binding domain [17,18,19]. The processing center cleaves both ends of the pri-miRNA and produces a two-nucleotide sticky end at the 3’ end [20,21]. DGCR8 is a nuclear and nucleolar-like protein with a size of approximately 90 bp and contains two dsRNA binding domains that bind to Drosha to recognize pri-miRNA. The core region of DGCR8 binds to heme to mediate dimers, and its interaction with ferric ions is required for the processing of pri-miRNA [22].

After processing by the microprocessor, exportin 5 (EXP5) transports pre-miRNA through the cell membrane to the cytoplasm for subsequent processing. EXP5 interacts with the GTP-binding protein (i.e., RanGTP) to form a glove-like spatial structure in which the positive charge carried by the internal space structure interacts with and tightly binds to the negative charge of the pre-miRNA. Finally, the pre-miRNA is transported from the nuclear pore to the cytoplasm. The inhibition of the EXP5 gene can lead to the decrease of miRNA expression level and the absence of pre-miRNA accumulation in the nucleus, which indicates that EXP5 is not only responsible for the transport of pre-miRNA during miRNA synthesis but also protects pre-miRNA from being decomposed by nuclease [23]. In addition to EXP5, there are other regulatory factors involved in the transport of miRNAs, such as exportin 1 (EXP1), which is sufficient to transport pre-miRNAs synthesized by the non-canonical pathway [24].

The pre-miRNA in the cytoplasm is digested by Dicer to cleave the stem-loop structure; this is then followed by the release of small RNA-dimers. First, the Dicer preferentially binds to two nucleotides protruding from the 3’ end of the pre-miRNA, and the Dicer usually cleaves from the 3’ end into a 22 bp small molecule RNA. At the same time, other regulatory factors are also involved in the regulation of Dicer processing, such as the transactivating response RNA binding protein (TRBP) traveling through the mitogen-activated protein kinase/ extracellular signal–regulated kinase (MAPK/ERK) signaling pathway to first regulate the processing of the human Dicer 1 enzyme before regulating miRNA organisms synthesis [25].

The small molecule RNA duplex obtained by Dicer processing binds to the Argonaute (AGO) protein to form a complex precursor (pri-miRISC) [26] which rapidly passes the miRNA passenger strand removal, the remaining mature guide strand, and the AGO protein to form a functional complex—namely the miRNA-induced silencing complex (miRISC) [27]. Finally, the miRNA can be combined with the target gene via a complementary pairing principle to degrade the corresponding target mRNA (full pairing) or inhibit its normal translation into the corresponding protein (partial pairing) to achieve the purpose of negatively regulating the target gene [28,29].

### 2.2. miRNA Expression Profiles in Ovarian Cancer 

Tumors constitute a major health concern and cause of death in humans. Their mechanisms of occurrence are complex with numerous influencing factors, with many tumor types exhibiting ready metastasis and recurrence. Moreover, many tumors are relatively difficult to treat and therefore have consistently been the focus and challenge of medical research. In recent years, the discovery of miRNA has provided new enlightenment to the study of tumors. In particular, various miRNAs have been shown to play different roles in ovarian cancer. For example, the expression of some miRNAs is inhibited in ovarian cancer, suggesting that these can be regarded as tumor suppressor genes. Conversely, other miRNAs are aberrantly expressed in ovarian cancer and can be regarded as cancer-promoting genes. By comparing the expression levels of miRNAs in ovarian cancer and normal tissues, Iorio et al. [30] found that the expression of miRNA-199a (miR-199a), miR-200a, miR-200b, and miR-200 was significantly higher than that in normal tissues, whereas miR-140, miR-145, and miR-125b1 showed low expression in cancer tissues. In turn, the miR-15 and miR-16 were down-regulated in ovarian cancer tissues, whereas miR-31 was expressed at low levels in serous ovarian cancer cells and tissues, suggesting its role as a tumor suppressor gene of ovarian cancer [31,32]. miR-200a, miR-200b, and miR-200c have also been shown to exhibit significantly higher levels in serous epithelial ovarian cancer than those in normal ovarian tissue [33]. In view of the varied expression and roles of different miRNAs in ovarian cancer, certain miRNAs can be considered as potential screening indicators to provide more specific molecular markers for the early diagnosis of ovarian cancer.

### 2.3. MicroRNA as Tumor Suppressors

A tumor suppressor gene is defined as a gene whose normal role serves to inhibit tumorigenesis, whereas its loss, mutation, or loss of function allows an activated oncogene to function and cause cancer. The product of the tumor suppressor gene may, for example, inhibit cell proliferation, promote cell differentiation, and/or inhibit cell migration, thereby playing a negative regulatory role. Notably, the expression levels of certain miRNAs in tumor tissues are significantly down-regulated, thus exhibiting the characteristics of tumor suppressor genes. 

In 2002, Calin et al. [34] first discovered the correlation between the abnormal expression of miRNA and tumors in a study of chronic lymphocytic leukemia (CLL). They found that miR-15 is involved in most CLL cells, whereas miR-16 expression was down-regulated or lacking. It was therefore speculated that miR-15 and miR-16 may play a role as tumor suppressor genes and be closely related to the occurrence of hematological tumors. Subsequently, Kumar et al. [35] confirmed that the expression of let-7 family miRNAs was significantly reduced in non-small cell lung cancer, thereby weakening the blocking effect on the gene for K-RAS and promoting the occurrence and developmental process of lung cancer. This indicated that the let-7 family of miRNAs also acts as tumor suppressor genes in non-small cell lung cancer. Moreover, miR-375 is associated with hepatocellular carcinoma, gastric cancer, esophageal squamous cell carcinoma, head and neck squamous cell carcinoma, cervical cancer, lung cancer, pancreatic cancer, and colorectal cancer. Particularly in melanoma, the expression of miR-375 is down-regulated to varying degrees, targeting and modulating important autologous oncogenes such as those encoding AEG-1, YAP1, IGF1R, and PDK1, which play a role in tumor suppression [28]. Luo et al. [36] found that miR-126 has a tumor suppressor-like effect in the ovarian cancer SKOV3 cell line, acting to inhibit ovarian cancer cells by inhibiting the expression of PAK4. Yamamoto et al. [37] found that, in endometrial cancer, the miR-1/miR-133a expression level is significantly down-regulated, whereas restoring its expression level can inhibit the metastasis and invasion of cancer cells by regulating phosphodiesterase 7A (PDE7A), suggesting the tumor suppressor-like role of this miRNA in the development of endometrial cancer. Cui et al. [38] found that miR-128 was significantly down-regulated in glioblastoma, with its down-regulated level being negatively correlated with the tumor stage. Moreover, Bmi-1, ARP5, and E2F-3a were also significantly up-regulated. ARP5 functions as a transcription inhibitor that can inhibit the expression of tumor suppressor genes. In turn, E2F-3a is a transcription factor involved in the control of the cell cycle. Further studies revealed that the introduction of exogenous miR-128 in the cell lines CRL-1690 and CRL-2610 can down-regulate the expression of Bmi-1, ARP5, and E2F-3a, suggesting that miR-128 is a potential tumor suppressor gene.

### 2.4. MicroRNAs as Oncogenes

Proto-oncogenes are generally genes involved in cell proliferation and are essential for maintaining normal life activities. The expression product of a proto-oncogene plays a precise regulatory role in the normal growth, reproduction, development, and differentiation of cells. The abnormal structures or uncontrolled expression of proto-oncogenes may lead to abnormal cell behaviors and the formation of tumors. The expression levels of certain miRNAs in tumor tissues are significantly up-regulated, exhibiting the characteristics of proto-oncogenes. Moreover, the up-regulation of their expression may lead to the malignant growth of cells. Iorio et al. [30] used gene chip technology to assess ovarian cancer tissues and found that miR-155 expression was up-regulated. The high expression of miR-155 also suggested poor prognosis in patients with lung cancer [39]. Iyevleva et al. [40] found that the expression of miR-21, miR-10b, and miR-31 was increased to different degrees in breast cancer—being closely related to the occurrence and development of breast cancer—and played a role similar to that of oncogenes. miR-103 is directly related to metastasis and poor prognosis in patients with colorectal cancer [30]. Martello et al. [41] have shown that high miR-107 expression levels in patients with breast cancer are closely related to patient low survival rate. Subsequently, the expression of miR-107 is significantly increased in many human tumors such as gastric, pancreatic, colorectal, and breast cancers, which suggests that it plays an important role in the development and progression of cancer [42].

## 3. Mechanism of miRNA in Ovarian Cancer Invasion and Metastasis

Ovarian cancer is one of the most common gynecological malignancies, and more than 90% of ovarian cancers are epithelial cancers. Though treatments and strategies continue to improve, the five-year survival rate for epithelial ovarian cancer is only 30%. In addition, ovarian cancer has no obvious symptoms in the early stage, and more than 60% of patients have abdominal or distant metastases when diagnosed with ovarian cancer [43,44]. miRNAs regulate the invasion and metastasis of ovarian cancer, and the abnormal expression of miRNA is closely related to the invasion and metastasis of ovarian cancer.

### 3.1. Ovarian Cancer Invasion and Metastasis

The invasion and metastasis of tumor cells can be divided into three independent stages. 1: Changes in tumor cell biochemistry, morphology, and migration ability (changes in stem cell-like properties and migration patterns); 2: Surface receptors that mediate metastasis; 3: The formation of a tumor microenvironment around the target organ is conducive to the survival and proliferation of metastatic cells [45,46]. The tumor microenvironment plays an important role in the development of tumors. Its hypoxia, low pH, and high-pressure environment is beneficial to the growth factors, proteolytic enzymes, and immunoinflammatory factors in the tumor cells. The processes of metastasis, differentiation, etc. have an important influence [47], and the abnormal expression of miRNA can change the ability of tumor invasion and metastasis by regulating the extracellular matrix [48].

### 3.2. miRNA and EMT

miR-200 and miR-429 are associated with the recurrence and survival rates of ovarian cancer, with their increased expression able to inhibit cancer metastasis [49]. Notably, the metastasis of cancer cells not only results in the poor prognosis of cancer treatment but also represents an important cause of cancer treatment failure. Epithelial-mesenchymal transition (EMT) is an important feature in tumor invasion and metastasis. In ovarian cancer, the progression of EMT is also confirmed to be closely related to tumor progression and metastasis. miR-125A induces hyperinvasiveness by reversing EMT in ovarian cancer cells [28]. Moreover, Li et al. [50] studied the expression of the EZRIN protein and related miRNAs in ovarian cancer and highly metastatic ovarian cancer by reverse transcription-polymerase chain reaction (PCR) and western blotting. They found that in low-metastatic ovarian cancer, the expression of miR-183 and miR-22 was higher than that in highly metastatic ovarian cancer. Increasing the expression of these miRNAs in ovarian cancer cells decreased the expression level of the EZRIN protein, suggesting that the inhibition of EZRIN through miR-183 and miR-22 as tumor suppressors plays a role in inhibiting ovarian cancer metastasis. 

### 3.3. Let-7 Family

miRNAs with highly homologous sequences are classified as a family, with the let-7 family being among the more widely studied miRNA groups. Let-7 exerts biological functions to inhibit cell proliferation and promote cell differentiation and apoptosis [51]. In different tumor tissues and cells, the let-7 family inhibits tumor cell growth and the invasion by inhibiting the expression of proto-oncogene-encoded proteins RAS, HMGA 2, c-Myc, c-Dc 25A, cdk 6, and cyclin 2 [51,52]. Let-7A-3/let-b is missing in 44% of ovarian cancer research samples, whereas restoring the expression of let-7b can significantly reduce the growth of ovarian tumors both in vitro and in vivo. In addition, let-7f is expressed at low levels in ovarian cancer cells with a high invasion and metastasis ability, suggesting that it may inhibit the invasion and metastasis of ovarian cancer [52]. miRNA families that can regulate tumor metastasis include the let-7 family (let-7/miR-48/84/241/265/793/794) [53], miR-10 family (miR-10a/b/c) [54], miR-16 family (miR-15/16/195/322/424/497) [55], miR-29 family (miR-29a/b/c/d) [56], miR-200 family (miR-200abc/429/548) [57], miR-221 family (mir-221/222), and miR-520 family (miR-93/105/106/291/294) /295/302/373/520). Among these, the miRNA families capable of regulating EMT include the let-7 family, the miR-16 family, the miR-200 family, and the miR-221 family. miRNAs are also abnormally expressed in cancer stem cells and can affect tumor formation and metastasis by regulating tumor stem cells. 

### 3.4. miR-200 Family

The miR-200 family (miR-200a, miR-141, miR-200b, miR-200c, miR-429) is a newly discovered tumor metastasis-associated miRNA. It inhibits the expression of the E-cadherin transcriptional inhibitors (zinc-finger enhancer binding 1) ZEB1 and ZEB2, increases the expression of E-cadherin, and promotes the transformation of mesenchymal cells into epithelial cells. The 3’UTR of ZEB1 and ZEB2 mRNA contains three and two miR-200a/141 potential binding sites, respectively. miR-200a binds to the 3’ UTR complementary series of ZEB1/ZEB2 mRNA via the 5’ end, resulting in the post-transcriptional inhibition of ZEB1/ZEB2 gene expression. In addition, the downregulation of miR-200a expression in tumor cells increased ZEB1/ZEB2 expression levels and inhibited E-cadherin. The cells were induced to undergo EMT to promote the invasion and metastasis of tumor cells; the up-regulation of miR-200a expression blocked EMT and inhibited the invasion and metastasis of tumor cells. However, ZEB1 and ZEB2 also inhibited the expression of the miR-200 family, partially reversing the mesenchymal-to-epithelial transition. Therefore, the ZEB gene family can be considered to form a dual negative regulatory pathway with the miR-200 family [58]. 

miR-429 belongs to one of the members of the miR-200 family, and the cells not only undergo morphological changes but also transform from mesenchymal cells to epithelial cells after the transfection of miR-429. EMT-related marker genes such as ZEB1, ZEB2, E-cadherin, N-cadherin, fibronectin 1 (FN1), and vimentin also change at mRNA levels [59]. Microarray detection also found that 296 genes were significantly down-regulated, such as the mesenchymal cell marker molecules ZEB1 and Versican (VCAN). Meanwhile, 373 genes were significantly up-regulated, including Tetraspanin 13 (TSPAN13), Caveolin 2 (CAV2), Desmoplakin (DSP), and the epithelium cell adhesion molecule (EPCAM). Therefore, miR-429 can be considered to play a key role in regulating EMT in ovarian cancer cells [60]. 

Thus, the miR-200 family differs in the expression of different tissues of ovarian cancer, at different stages, and in benign or malignant ovarian cancer. This suggests that it may induce ovarian cancer as a proto-oncogene, and its involvement in ovarian cancer invasion and metastasis is closely related to a poor prognosis. miR-200 has important molecular markers as a potential target for ovarian cancer invasion and metastasis, as well as the prognosis and treatment of ovarian cancer metastasis.

### 3.5. miRNAs Regulate Expression and Function of Extracellular Matrix

The extracellular matrix constitutes a dynamic system of cellular mechanical changes and biochemical changes in extracellular fluid. The composition between cells and fibers includes proteoglycans, polyglycoproteins, and glycosaminoglycans; Extracellular matrix (ECM) molecules regulate cell migration and tumor metastasis [61]. The down-regulation of miRNAs regulates tumor metastasis by degrading ECM. [62]. The matrix metalloproteinases (MMPs) family is a class of zinc-dependent endopeptidases that degrade and remodel ECM to accelerate tumor metastasis [63]. MMPs are important proteases in tumor invasion. MMP-2, MMP-3, MMP-7, and MMP-9 are thought to be involved in the invasion and migration of tumor cells in ovarian cancer. By comparing ovarian cancer tissues with distant non-ovarian cancer tissues in ovarian cancer patients, MMP-7 was significantly increased in ovarian cancer tissues, and miR-543 was significantly reduced. miR-543 inhibits MMP-7 transcription by binding to the 3’UTRs of MMP-7 mRNA, and it reduces the metastasis of ovarian cancer cells [64]. In addition, miR-205 showed a dramatic increase in ovarian cancer in the late stage. Since miR-205 inhibits transcription factor-21 (TCF-21), it increases the ability of ovarian cancer cells to spread and metastasize [65]. In addition, the invasiveness of miR-205-transfected ovarian cancer cells was significantly increased five-fold. Conversely, the inhibition of miR-205 reduced cell invasiveness. This suggests that miR-205 plays a role in proto-oncogenes and promotes tumor invasion during ovarian cancer. 

### 3.6. miRNA in Angiogenesis

Tumor angiogenesis is an important process of tumor progression which facilitates the dissemination and metastasis of tumor cells through blood circulation [45]. The miR-200 family is not only associated with EMT but also with tumor angiogenesis. It has been reported that the miR-200 family inhibits the formation of blood vessels by directly or indirectly targeting the interleukin-8 secreted by tumor epithelial cells and the chemokine CXCL-1. Transferring the miR-200 family to the tumor epithelium showed a significant decrease in tumor cell metastasis and angiogenesis, as well as normalization of blood vessels [66]. Vascular endothelial growth factor (VEGF) also plays an important role in promoting the invasion and metastasis of ovarian cancer. As a pro-angiogenic factor, it also promotes endothelial cell growth and migration while increasing vascular permeability. Moreover, VEGF also increases the expression of miR-205 in ovarian cancer cells, reduces ezrin and lamin A/C, and promotes the proliferation and metastasis of ovarian cancer cells [67].

## 4. Use of MicroRNAs in Diagnosis

Differences between the miRNA profiles of ovarian cancer and the potential role of miRNAs in ovarian cancer diagnosis have been assessed in several studies (Table 1). The diagnosis of ovarian cancer has always been a significant problem, as approximately 70–80% of patients present with advanced cancer at diagnosis. However, the results of miRNA research have brought new methods to the diagnosis of ovarian cancer [68]. Studies, including one by Resnick et al. [69], have confirmed that miRNA in serum can be used as a marker for ovarian cancer. The investigators selected serum samples from 28 confirmed but untreated patients with ovarian cancer patients, along with 15 normal human serum samples as controls, and subjected samples from nine of the patients with ovarian cancer to fluorescence array analysis using human miRNA panels along with real-time PCR. Analysis of miRNAs in these patients and four normal human serum samples revealed that 21 miRNAs exhibited differences in expression between ovarian cancer and normal human serum. Subsequent detection of these 21 miRNAs in the remaining 19 ovarian cancer and 11 normal serum samples using real-time quantitative PCR confirmed eight miRNAs as being differentially expressed between ovarian cancer and normal samples. Up-regulated miRNAs included miR-21, miR-92, miR-29a, miR-93, and miR-126, whereas the down-regulated miRNAs were miR-99b, miR-127, and miR-155. The results of this study fully demonstrate the diagnostic significance of serum miRNA for ovarian cancer. 

Furthermore, Zhou et al. [70] and other studies found that miRNA in urine also has important significance in the diagnosis of ovarian cancer. The researchers collected and compared urinary specimens from 39 patients with ovarian serous adenocarcinoma, 26 patients with benign gynecological disease, and 30 healthy controls to study the clinical value of miRNA in urine for detecting ovarian serous adenocarcinoma. miRNA microarray data showed that the expression of miR30a-5p was significantly up-regulated in patients with ovarian serous adenocarcinoma compared with that in the benign gynecological disease and healthy control groups. In addition, the researchers also examined miRNA expression in the urine of the subset of patients who underwent tumor resection. Notably, the expression of miR-30a-5p was significantly decreased, suggesting that urinary miR-30a-5p is derived from the ovarian serous adenocarcinoma itself. Therefore, the expression level of miR-30a-5p in urine may be used as an important basis for the diagnosis of ovarian serous adenocarcinoma.

## 5. miRNAs as Therapeutic Targets

Ovarian cancer treatment is based on surgery-based comprehensive treatment including chemotherapy as an important component; however, chemotherapy resistance directly affects treatment efficacy and patient prognosis. An in-depth study of miRNA is expected to provide new evidence and solutions for chemotherapy resistance. Yang et al. [79] found that the expression of miR-214 and miR-150 is abnormally up-regulated in ovarian cancer, which can inhibit the target gene *PTEN*, down-regulate the PTEN protein, and initiate the protein kinase B (AKT) pathway. Conversely, AKT blockage or preventing miRNA-*PTEN* interaction can inhibit ovarian cancer, apoptosis, the induction of cancer cell proliferation, and chemotherapeutic drug tolerance. This suggests that miR-214 and miR-150 play important roles in ovarian cancer, whereas their reduction would achieve the purpose of adjuvant ovarian cancer treatment [80]. Lan et al. [81] found that the expression of miR-140-5P in patients with ovarian cancer was significantly decreased, negatively regulating its target gene *PDGFRA*. Conversely, increasing miR-140-5P expression promoted the proliferation and migration of ovarian cancer cells. The study also showed that miR-140-5P down-regulation could reduce the sensitivity of ovarian cancer cells to cisplatin. These findings indicate that miR-140-5P might serve as another important potential target for ovarian cancer treatment; however, up-regulation rather than inhibition would be required to inhibit the proliferation and migration of ovarian cancer cells and enhance their sensitivity to cisplatin, thereby achieving the therapeutic purpose of inhibiting the progression of ovarian cancer. 

In addition, Echevarria-Vargas et al. [82] found that miR-21 is also associated with cisplatin resistance in ovarian cancer. The detection of miR-21 by reverse transcription-PCR revealed miR-21 the down-regulation in cisplatin-sensitive compared to cisplatin-resistant ovarian cancer. Notably, whereas increasing the expression level of miR-21 in cisplatin-sensitive cancer cells increased the proliferation of ovarian cancer cells, targeting miR-21 reduced the tumorigenic characteristics of the cisplatin-resistant cells. Moreover, Aqeilan et al. [19] found that miR-15 and miR-16 are involved in the multi-drug resistance of cells by acting on the target gene encoding Bcl-2. Dai et al. [83] re-expressed the *PTEN* gene concomitant with ovarian cancer suppression by targeting miR-29a to cancer tissues. The potential anti-tumor effects following miR-29a transfection were inferred through downstream molecular expression and the apoptosis of ovarian cancer cells. Lu et al. [51] observed that the expression of let-7a was significantly lower in patients with ovarian cancer who were sensitive to platinum and paclitaxel compared with those who were resistant to those agents; on the other hand, the high expression of let-7a was associated with an increased survival rate in patients receiving chemotherapy based on platinum alone but a lower survival in those receiving combination therapy. These findings suggest that let-7a can be used as a potential biomarker to determine the response of patients with ovarian cancer to chemotherapy. 

However, studies have demonstrated that tumor cells can simultaneously exhibit the dysregulation of multiple miRNAs, suggesting that treatment of individual miRNAs may not be sufficient. Therapy targeting the transcription of multiple miRNAs—one that uses, for example, an anti-microRNA antisense oligodeoxyribonucleotide—may thus be required to simultaneously counter amassed miRNA dysregulation [84].

## 6. Prognosis of miRNA and Ovarian Cancer

Numerous studies have shown that miRNA can be used to predict the prognosis of patients with ovarian cancer [85]. Gao et al. [86] collected serum samples from 74 patients with epithelial ovarian cancer, 19 patients with borderline ovarian cancer, and 50 healthy controls, and they determined the levels of miR-200c and miR-142 by quantitative PCR. In addition, their expression was evaluated by the Kaplan–Meier curve and log-rank test to evaluate the prognostic value for ovarian cancer. The results showed that the expression of these miRNAs in the serum of patients with ovarian cancer was significantly higher than that of the healthy control group. The relative expression level of serum miR-200c revealed a downward trend from early to advanced ovarian cancer, with the two-year survival rate of patients with relatively high expression of miR-200c being higher than that of patients with relatively low expression. In comparison, miR-141 exhibited the opposite pattern.

## 7. Future Perspectives

Ovarian cancer, as a common gynecological malignancy with high mortality, offers considerable challenges for early diagnosis owing to the lack of obvious symptoms until the tumor has metastasized. However, as for other types of tumors, miRNAs show promise as stable and powerful markers closely related to ovarian tumor status and have become a hot spot in the field of medical tumor research. Nevertheless, owing to their relatively recent discovery, research regarding miRNA regulatory mechanisms and target genes is still in its infancy, and their relationship with tumors and specific tissues, especially the ovaries, is incompletely understood. Therefore, research regarding the relationship between miRNAs and cancer needs to be further deepened and expanded. The authors believe that the constantly updated research results will facilitate in-depth analyses of the mechanism of ovarian cancer development, thereby allowing more effective guidance of clinicians for the early diagnosis of patients with ovarian cancer, along with the individualization of treatment and improved scientific assessment of prognosis to the ultimate benefit of patients with ovarian cancer.

## Figures and Tables

**Figure 1 ijerph-16-01510-f001:**
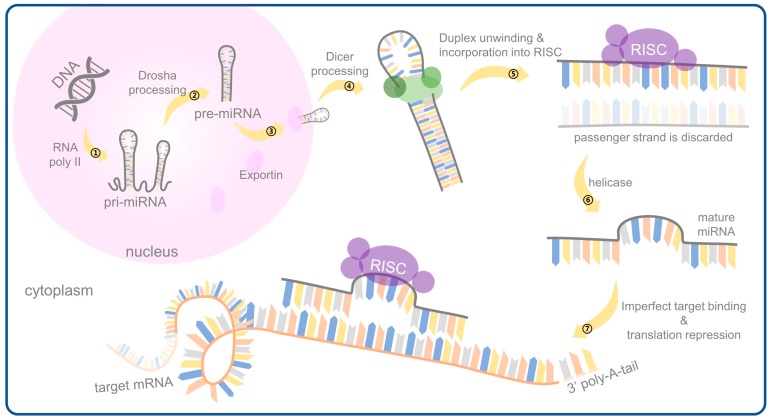
An overview of miRNA biogenesis and effector pathways. The transcription of a primary miRNA (pri-miRNA) by RNA polymerase II is followed by its processing by Drosha in complex into a precursor hairpin (pre-miRNA); these steps occur in the nucleus. Following transportation by Exportin from nucleus to cytoplasm, the pre-miRNAs are processed by Dicer. A miRNA-loaded RNA-induced silencing complex (RISC) mediates gene silencing via mRNA cleavage and degradation or translational repression, depending on the complementarity between the miRNA and the targeted mRNA transcript. The result is either the inhibition of translation or the degradation of the mRNA target, depending on incomplete or complete complementarity to the target mRNA sequence.

**Table 1 ijerph-16-01510-t001:** miRNAs as diagnostic markers in ovarian cancer.

Specimen	Alterations	miRNA	Ref.
**Serous carcinoma**	↑	miR-205, miR-429, miR-141, miR-200c, miR-93, miR-16, miR-20a, miR-21, miR-27a, miR-200a, miR-200b, miR-200c	[33,71,72]
	↓	miR-320c, miR-383, let-7b, miR-99a, miR-125b, miR-145, miR-100, miR-31, miR-137, miR-132, miR-26a	[71,72,73]
**EOC cell line**	↑	miR-26, miR-26b, miR-103, miR-182, miR-203	[74,75]
	↓	miR-377, miR-432, miR-124a, miR-436, let-7d	[74]
**Clear cell carcinoma**	↑	miR-93, miR-126, miR-338-3p, miR-200a, miR-200c, miR-30a, miR-141, miR-182-5p, miR-200a-3p, miR-510	[69,72,76,77]
	↓	miR-383, miR-424-5p, miR-127, miR-155, miR-99b	[69,72,77]
**Endometrioid carcinoma**	↑	miR-21, miR-29a, miR-92, miR-30c1, miR-126	[69]
	↓	miR-342-3p, miR-181a-3p, miR-450b-5p, miR-155, miR-127, miR-99b	[69,78]

EOC: Epithelial ovarian cancer.

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
