# Peer review of "MicroRNA in Ovarian Cancer: Biology, Pathogenesis, and Therapeutic Opportunities"

_ijerph, 2019, doi:10.3390/ijerph16091510_

Round 1

Reviewer 1 Report

This is an interesting review describing the expression of miRNA molecules in ovarian cancer and summarises data relating to their functionality in this disease. It reads well and my view is that it is well worth publishing.

Minor suggestions

Abstract Line 7: I was unclear as to the meaning of : “By pairing with a thiol specific to a target messenger RNA..” ?   What does the “thiol” refer to?  I’m aware that chemically modified molecules have incorporated thiol groups but is this present in the natural molecule? If so, could a reference please be supplied..

Page 2 Line 48:  “with metastasis and invading cancer cells” rather than “with metastasized and invaded cancer cells”

Page 2 Line 53: “the complete remission rate can reach 70-80%” rather than “the complete remission can reach 70-80%”

Page 2 Line 55: Unsure what is meant by “transfer”? If this word is removed, the sentence still makes sense 

Page 5 Line 205: “may lead to” rather than “may to”

Page 5 Line 220: “has always been a significant problem” rather than “has always been an academic problem”

Page 6 Line 257: “play important roles” rather than “play an important role “

Page 7 Line 263: “as another” rather than “asanother”

In Table 1, I am unclear as to why “EOC cell line” has been included alongside the major histologies (serous, clear cell, endometrioid carcinomas.” ?

Author Response

Dear referee 1:

We would like to thank the Reviewers for his/her kind expression of our original work. In addition, all the authors are truly appreciative for the Reviewers’ effort, and thank the Reviewer s’ for his/her critical and highly constructive comments on how to improve this manuscript. We are grateful for the opportunity to revise our manuscript; we have endeavored to address the problems indicated by the Reviewer.

Comments and Suggestions for Authors

This is an interesting review describing the expression of miRNA molecules in ovarian cancer and summarises data relating to their functionality in this disease. It reads well and my view is that it is well worth publishing.

Minor suggestions

Abstract Line 7: I was unclear as to the meaning of : “By pairing with a thiol specific to a target messenger RNA..” ?   What does the “thiol” refer to?  I’m aware that chemically modified molecules have incorporated thiol groups but is this present in the natural molecule? If so, could a reference please be supplied..

Page 2 Line 48:  “with metastasis and invading cancer cells” rather than “with metastasized and invaded cancer cells”

Page 2 Line 53: “the complete remission rate can reach 70-80%” rather than “the complete remission can reach 70-80%”

Page 2 Line 55: Unsure what is meant by “transfer”? If this word is removed, the sentence still makes sense 

Page 5 Line 205: “may lead to” rather than “may to”

Page 5 Line 220: “has always been a significant problem” rather than “has always been an academic problem”

Page 6 Line 257: “play important roles” rather than “play an important role “

Page 7 Line 263: “as another” rather than “asanother”

In Table 1, I am unclear as to why “EOC cell line” has been included alongside the major histologies (serous, clear cell, endometrioid carcinomas.” ?

Response: We thank the reviewer for checking our article in detail and indicating the unclear information. We have corrected the mistake. We use red letters to indicate where the changes have been made.

Reviewer 2 Report

The authors have a contributed a thoughtful manuscript outlining the role of miRNA's in the diagnosis, contribution and amelioration of ovarian cancer. The manuscript is written with a sense of technical proficiency but the text wanders with keeping in intention (ovarian cancer) set out in title and abstract.  Also the role of miRNAs mostly reads like a grocery/laundry list of differentiatial expression and could use some thoughtful synthesis of the actually biological pathways regulated by RNAs. Specific examples are given below. With additional work this could be an excellent resource for others. 

Specific comments below

The section 107-142 speaks too much in sweeping broad generalities.  Where possible speak of specific pathways and genes controlled by particular mRNA in ovarian cancer. We need more biology than simply pro-or anti tumor progression.   If readers have to go back to the primary resources to find the most pertinent information then the review is not successful in accomplishing its mission in making information accessible. It is not news or even necessary to comment that some miRNAs are different between pathological and healthy conditions, this is expected. There is nothing wrong with providing the list of different miRNA's but we should be brought up to date on their targets and how those targets are proposed to contribute to disease.

107-125 has broad concepts reiterated twice. The first time with a complete lack of detail (miRNAs are different in health and disease) and the second time miRNAs are different in health and disease and here's a list of different ones. This serves as a good example of how to consolidate concepts and keep focus in a review.  

Similar in 158-163 "can affect tumor formation and 158 metastasis by regulating tumor stem cells. This ability may be due to their important role in the 159 development and proliferation of embryonic and normal stem cells; for example, miR-145 can 160 regulate lung cancer stem cell function and tumors. In the EMT process, miR-200c inhibits the 161 formation of mammary gland and breast cancer stem cell proliferation and the tumorigenic ability of 162 breast stem cells."

HOW do these microRNAs affect these processes? Its not simply they are expressed and magic happens, there are biological pathways these regulate the make since in terms of how they impact development and proliferation and the stem cell niche.  Some details need to be add for these words to be impactful and worth including.

L. 174 I believe miR1 is supposed to be miRNA15. Please check

179-184. These sentences don't naturally read in a related way to one another. One sentence is about every cancer other than melanoma and the next sentence is only about melanoma. It is unclear if the targets of miRNA in melanoma are also the same as in the previous sentence. 

Line 185 - Authors comment about the confirmation of a role of miR-126 without talking about a role for mRNA126 prior or in a related context. 

Line 189, for key targets like PDE7a what is the biological function of these molecules and how do they contribute to disease.

Line 219 The diagnosis of ovarian 219 cancer has always been an academic problem as approximately 70% of patients present with 220 advanced cancer at diagnosis.

How is this an academic problem?

In general, the manuscript, while reflecting professional competency, reads mostly like a laundry list of differential expression of miRNAs and fluctuates between different types of tumors without a logical justification for doing so.  If studies in other tumor confirm the essential biology of targeted miRNA interaction then they reinforce findings in ovarian cancer and that biology and that association should be discussed.  

I believe the authors have all the references at their disposal to generate an excellent review and are 80% of the way there. Additional biological detail (some examples outline above) with a focused attention to the point of every paragraph in a particular as it related to ovarian cancer will hone the manuscript to a more useful state. 

I encourage the authors to pursue this further and resubmit what will undoubtedly be a fine review.

Author Response

Dear referee 2:

We would like to thank the Reviewers for his/her kind expression of our original work. In addition, all the authors are truly appreciative for the Reviewers’ effort, and thank the Reviewer s’ for his/her critical and highly constructive comments on how to improve this manuscript. We are grateful for the opportunity to revise our manuscript; we have endeavored to address the problems indicated by the Reviewer.

Comments and Suggestions for Authors

The authors have a contributed a thoughtful manuscript outlining the role of miRNA's in the diagnosis, contribution and amelioration of ovarian cancer. The manuscript is written with a sense of technical proficiency but the text wanders with keeping in intention (ovarian cancer) set out in title and abstract.  Also the role of miRNAs mostly reads like a grocery/laundry list of differentiatial expression and could use some thoughtful synthesis of the actually biological pathways regulated by RNAs. Specific examples are given below. With additional work this could be an excellent resource for others. 

Specific comments below

The section 107-142 speaks too much in sweeping broad generalities.  Where possible speak of specific pathways and genes controlled by particular mRNA in ovarian cancer. We need more biology than simply pro-or anti tumor progression.   If readers have to go back to the primary resources to find the most pertinent information then the review is not successful in accomplishing its mission in making information accessible. It is not news or even necessary to comment that some miRNAs are different between pathological and healthy conditions, this is expected. There is nothing wrong with providing the list of different miRNA's but we should be brought up to date on their targets and how those targets are proposed to contribute to disease.

107-125 has broad concepts reiterated twice. The first time with a complete lack of detail (miRNAs are different in health and disease) and the second time miRNAs are different in health and disease and here's a list of different ones. This serves as a good example of how to consolidate concepts and keep focus in a review.  

Similar in 158-163 "can affect tumor formation and 158 metastasis by regulating tumor stem cells. This ability may be due to their important role in the 159 development and proliferation of embryonic and normal stem cells; for example, miR-145 can 160 regulate lung cancer stem cell function and tumors. In the EMT process, miR-200c inhibits the 161 formation of mammary gland and breast cancer stem cell proliferation and the tumorigenic ability of 162 breast stem cells."

HOW do these microRNAs affect these processes? Its not simply they are expressed and magic happens, there are biological pathways these regulate the make since in terms of how they impact development and proliferation and the stem cell niche.  Some details need to be add for these words to be impactful and worth including.

Response: We thank the Reviewer for providing the worthwhile suggestion on improving the quality of our article. We attempted to increase the mechanism of action of MiRNA in ovarian cancer invasion and metastasis in the third section. (line 198-306)

“ 3. Mechanism of MiRNA in Ovarian Cancer Invasion and Metastasis

Ovarian cancer is one of the most common gynecological malignancies, and more than 90% of ovarian cancers are epithelial cancers. Although treatments and strategies continue to improve, the 5-year survival rate for epithelial ovarian cancer is only 30%. In addition, ovarian cancer has no obvious symptoms in the early stage, and more than 60% of patients have abdominal or distant metastases when diagnosed with ovarian cancer [41, 42]. The process of miRNA and ovarian cancer invasion and metastasis has been highly valued, and the abnormal expression of miRNA is closely related to the invasion and metastasis of ovarian cancer.

3.1 Ovarian cancer invasion and metastasis

The invasion and metastasis of tumor cells can be divided into three independent stages: 1. Changes in tumor cell biochemistry, morphology and migration ability (changes in stem cell-like properties and migration patterns); 2. Surface receptors that mediate metastasis; The formation of tumor microenvironment around the target organ is conducive to the survival and proliferation of metastatic cells [43]. The tumor microenvironment plays an important role in the development of tumors. Its hypoxia, low pH and high pressure environment are beneficial to the growth factors, proteolytic enzymes and immunoinflammatory factors in the tumor cells. The process of metastasis, differentiation, etc. has an important influence [44], and the abnormal expression of miRNA can change the ability of tumor invasion and metastasis by regulating the extracellular matrix [45].

3.3. MiR-200 Family

The miR-200 family (miR-200a, miR-141, miR-200b, miR-200c, miR-429) is a newly discovered tumor metastasis-associated miRNA. It inhibits the expression of E-cadherin transcriptional inhibitors (zinc-finger enhancer binding 1) ZEB1 and ZEB2, increases the expression of E-cadherin, and promotes the transformation of mesenchymal cells into epithelial cells. The 3'UTR of ZEB1 and ZEB2 mRNA contains 3 and 2 miR-200a/141 potential binding sites, respectively. MiR-200a binds to the 3' UTR complementary series of ZEB1/ZEB2 mRNA via the 5' end, resulting in post-transcriptional inhibition of ZEB1/ZEB2 gene expression. In addition, downregulation of miR-200a expression in tumor cells increased ZEB1/ZEB2 expression levels and inhibited E-cadherin. The cells were induced to undergo EMT to promote invasion and metastasis of tumor cells; up-regulation of miR-200a expression blocked EMT and inhibited invasion and metastasis of tumor cells. However, ZEB1 and ZEB2 also inhibited the expression of the miR-200 family, partially reversing the mesenchymal–to-epithelial transition. Therefore, the ZEB gene family can be considered to form a dual negative regulatory pathway with the miR-200 family [55].

miR-429 belongs to one of the members of the miR-200 family, and the cells not only undergo morphological changes, but also transform from mesenchymal cells to epithelial cells after transfection of miR-429. EMT-related marker genes such as ZEB1, ZEB2, E-cadherin, N-cadherin, fibronectin 1 (FN1), and vimentin also change at mRNA levels [56]. Microarray detection also found that 296 genes were significantly down-regulated, such as the mesenchymal cell marker molecules ZEB1 and Versican (VCAN); while 373 genes were significantly up-regulated, including Tetraspanin 13 (TSPAN13), Caveolin 2 (CAV2), Desmoplakin (DSP), epithelium cell adhesion molecule (EPCAM). Therefore, miR-429 can be considered to play a key role in regulating EMT in ovarian cancer cells [57].

Thus, the miR-200 family differs in the expression of different tissues of ovarian cancer, at different stages, and in benign or malignant ovarian cancer. This suggests that it may induce ovarian cancer as a proto-oncogene, and involvement in ovarian cancer invasion and metastasis is closely related to poor prognosis. miR-200 has important molecular markers as a potential target for ovarian cancer invasion and metastasis and prognosis and treatment of ovarian cancer metastasis.

3.4. MiRNAs Regulate Expression and Function of Extracellular Matrix

The extracellular matrix constitutes a dynamic system of cellular mechanical changes and biochemical changes in extracellular fluid. The composition between cells and fibers includes proteoglycans, polyglycoproteins and glycosaminoglycans, and ECM molecules regulate cell migration and tumor metastasis[58]. Abnormal expression of miRNA can regulate the ability of tumor metastasis by degrading ECM [59]. The matrix metalloproteinases (MMPs) family is a class of zinc-dependent endopeptidases that degrade and remodel ECM to accelerate tumor metastasis [60]. MMPs are important proteases in tumor invasion. MMP-2, MMP-3, MMP-7 and MMP-9 are thought to be involved in the invasion and migration of tumor cells in ovarian cancer. In ovarian cancer patients, by comparing ovarian cancer tissues with distant non-ovarian cancer tissues, MMP-7 was significantly increased in ovarian cancer tissues, and miR-543 was significantly reduced. miR-543 inhibits MMP-7 transcription by binding to the 3'UTRs of MMP-7 mRNA and reduces metastasis of ovarian cancer cells [61]. In addition, miR-205 showed a dramatic increase in ovarian cancer in the late stage. Since miR-205 inhibits transcription factor-21 (TCF-21), it increases the ability of ovarian cancer cells to spread and metastasize [62]. In addition, the invasiveness of miR-205-transfected ovarian cancer cells was significantly increased 5-fold. Conversely, inhibition of miR-205 reduced cell invasiveness. This suggests that miR-205 plays a role in proto-oncogenes and promotes tumor invasion during ovarian cancer.

3.5. MiRNA in Angiogenesis

Tumor angiogenesis is an important process of tumor progression, which facilitates the dissemination and metastasis of tumor cells through blood circulation [43]. The miR-200 family is not only associated with EMT but also with tumor angiogenesis. It has been reported that the miR-200 family inhibits the formation of blood vessels by directly or indirectly targeting the interleukin-8 secreted by tumor epithelial cells and the chemokine CXCL-1. Transferring the miR-200 family to the tumor epithelium showed a significant decrease in tumor cell metastasis and angiogenesis, as well as normalization of blood vessels [63]. Vascular endothelial growth factor (VEGF) also plays an important role in promoting the invasion and metastasis of ovarian cancer. As a pro-angiogenic factor, it also promotes endothelial cell growth and migration while increasing vascular permeability. Moreover, VEGF also increases the expression of miR-205 in ovarian cancer cells, reduces Ezrin and lamin A/C, and promotes proliferation and metastasis of ovarian cancer cells[64].”

L. 174 I believe miR1 is supposed to be miRNA15. Please check

Response: We thank the reviewer for pointing out the inappropriate description. We have corrected the mistake. (line 156)

179-184. These sentences don't naturally read in a related way to one another. One sentence is about every cancer other than melanoma and the next sentence is only about melanoma. It is unclear if the targets of miRNA in melanoma are also the same as in the previous sentence. 

Response: All authors thank the Reviewer for indicating the unclear point and giving us opportunity to clarify it. We have revised the described. (line 161-166)

Line 185 - Authors comment about the confirmation of a role of miR-126 without talking about a role for mRNA126 prior or in a related context. 

Response: All authors thank the Reviewer for indicating the unclear point and giving us opportunity to clarify it. We have revised the described. (line 166)

Line 189, for key targets like PDE7a what is the biological function of these molecules and how do they contribute to disease.

Response: All authors thank the Reviewer for indicating the unclear point and giving us opportunity to clarify it. (line 170)

“ The protein encoded by this gene belongs to the cyclic nucleotide phosphodiesterase (PDE) family, and PDE7 subfamily. This PDE hydrolyzes the second messenger, cAMP, which is a regulator and mediator of a number of cellular responses to extracellular signals. Thus, by regulating the cellular concentration of cAMP, this protein plays a key role in many important physiological processes. “

Line 219 The diagnosis of ovarian 219 cancer has always been an academic problem as approximately 70% of patients present with 220 advanced cancer at diagnosis.

How is this an academic problem?

Response: We thank the reviewer for pointing out the inappropriate description. We have corrected the mistake. (line 309)

In general, the manuscript, while reflecting professional competency, reads mostly like a laundry list of differential expression of miRNAs and fluctuates between different types of tumors without a logical justification for doing so.  If studies in other tumor confirm the essential biology of targeted miRNA interaction then they reinforce findings in ovarian cancer and that biology and that association should be discussed.  

I believe the authors have all the references at their disposal to generate an excellent review and are 80% of the way there. Additional biological detail (some examples outline above) with a focused attention to the point of every paragraph in a particular as it related to ovarian cancer will hone the manuscript to a more useful state. 

I encourage the authors to pursue this further and resubmit what will undoubtedly be a fine review.

Response: We thank the reviewer for giving advice. We have reorganized the manuscript and added descriptions related to ovarian cancer to try to improve the quality of the article.

Reviewer 3 Report

This manuscript provides a succinct description of miRNAs and their roles in ovarian cancer. In particular, the authors review clinical information regarding miRNAs as diagnostic and therapeutic factors. Overall, this manuscript is well-written and provides some interesting points regarding miRNA as disease biomarkers. However, there are several critical points that require further attention before publication:

In the abstract and in lines 78/127 the authors refer to miRNA:mRNA interactions involving thiol groups. This is incorrect and I believe the authors mean phosphate groups. Regardless, this should be corrected and in a more detailed description of the molecular structural characteristics of miRNA should be provided with appropriate citations. 

The miRNA biogenesis section is lacking detail. For example, the article and Figure 1 alludes to polII-mediated miRNA transcription. However, polIII also has a large role in miRNA transcription. Thus, a better description of miRNA biogenesis is needed.

The section of miRNA expression profiles (lines 106 to 125) is too general and not entirely informative. The authors basically describe the concept of increased and decreased miRNA expression patterns without providing relevant information in an organized manner. Therefore, this section needs to be better organized by discussing miRNAs commonly associated OVCa (in a table) and provide a discussion of the samples and detection platforms used in these studies.

A few spelling errors throughout and minor editing.

Author Response

Dear referee 3:

We would like to thank the Reviewers for his/her kind expression of our original work. In addition, all the authors are truly appreciative for the Reviewers’ effort, and thank the Reviewer s’ for his/her critical and highly constructive comments on how to improve this manuscript. We are grateful for the opportunity to revise our manuscript; we have endeavored to address the problems indicated by the Reviewer.

Comments and Suggestions for Authors

This manuscript provides a succinct description of miRNAs and their roles in ovarian cancer. In particular, the authors review clinical information regarding miRNAs as diagnostic and therapeutic factors. Overall, this manuscript is well-written and provides some interesting points regarding miRNA as disease biomarkers. However, there are several critical points that require further attention before publication:

In the abstract and in lines 78/127 the authors refer to miRNA:mRNA interactions involving thiol groups. This is incorrect and I believe the authors mean phosphate groups. Regardless, this should be corrected and in a more detailed description of the molecular structural characteristics of miRNA should be provided with appropriate citations. 

Response: We thank the reviewer for checking our article in detail and indicating the unclear information. We have corrected the mistake. We use red letters to indicate where the changes have been made.

The miRNA biogenesis section is lacking detail. For example, the article and Figure 1 alludes to polII-mediated miRNA transcription. However, polIII also has a large role in miRNA transcription. Thus, a better description of miRNA biogenesis is needed.

Response: We thank the Reviewer for providing the worthwhile suggestion on improving the quality of our article. We attempted to increase the mechanism of action of MiRNA synthesis in ovarian cancer invasion and metastasis in 2.1.2. section. (line 78-111)

“ 2.1.2 MiRNA Synthesis:

The synthesis and processing of miRNAs are illustrated in Fig. 1.

The pri-miRNA is processed by a microprocessor to form a 60-70 bp hairpin-like small RNA (ie, pre-miRNA precursor) in the nucleus. The microprocessor is composed of ribonuclease Drosha and microprocessor complex subunit 8 (DGCR8) constitutes a complex [14]. Like Dicer, Drosha belongs to the RNA polymerase III family and is a class of endonucleases that act on double-stranded RNA (dsRNA). Drosha is a processing center consisting of two RNA polymerase III domains and a dsRNA binding domain [15-17]. The processing center cleaves both ends of the pri-miRNA and produces a 2 nucleotide sticky end at the 3' end [18, 19]. DGCR8 is a nuclear and nucleolar-like protein with a size of approximately 90 ku and contains two dsRNA binding domains that bind to Drosha to recognize pri-miRNA. The core region of DGCR8 binds to heme to mediate dimers, and its interaction with ferric ions is required for the processing of pri-miRNA [20].

After processing by microprocessor, exportin 5 (EXP5) transports pre-miRNA through the cell membrane to the cytoplasm for subsequent processing and processing. EXP5 interacts with the GTP-binding protein (ie, RanGTP) to form a glove-like spatial structure in which the positive charge carried by the internal space structure interacts with and tightly binds to the negative charge of the Pre-miRNA. Finally, the pre-miRNA is transported from the nuclear pore to the cytoplasm. The inhibition of EXP5 gene can lead to the decrease of miRNA expression level and the absence of pre-miRNA accumulation in the nucleus, which indicates that EXP5 is not only responsible for the transport of pre-miRNA during miRNA synthesis, but also protects pre-miRNA from being decomposed by nuclease [21]. In addition to EXP5, there are other regulatory factors involved in the transport of miRNAs, such as exportin 1 (EXP1), which is sufficient to transport Pre-miRNAs synthesized by the non-canonical pathway [22].

The Pre-miRNA in the cytoplasm is digested by Dicer to cleave the stem-loop structure, followed by the release of small RNA-dimers. First, the Dicer preferentially binds to two nucleotides protruding from the 3' end of the pre-miRNA, and the Dicer usually cleaves from the 3' end into a 22 bp small molecule RNA. At the same time, other regulatory factors are also involved in the regulation of Dicer processing, such as transactivating response RNA binding protein (TRBP) through the MAPK/ERK signaling pathway to regulate the processing of human Dicer 1 enzyme, and then regulate miRNA organisms Synthesis [23].

The small molecule RNA duplex obtained by Dicer processing binds to the Argonaute (AGO) protein to form a complex precursor (pri-miRISC) [24], which rapidly passes the miRNA passenger strand removal, the remaining mature guide strand and AGO protein then form a functional complex, namely miRNA-induced silencing complex (miRISC) [25]. Finally, the miRNA can be combined with the target gene via a complementary pairing principle to degrade the corresponding target mRNA (full pairing) or inhibit its normal translation into the corresponding protein (partial pairing) to achieve the purpose of negatively regulating the target gene [26, 27]. “

The section of miRNA expression profiles (lines 106 to 125) is too general and not entirely informative. The authors basically describe the concept of increased and decreased miRNA expression patterns without providing relevant information in an organized manner. Therefore, this section needs to be better organized by discussing miRNAs commonly associated OVCa (in a table) and provide a discussion of the samples and detection platforms used in these studies.

A few spelling errors throughout and minor editing.

Response: We thank the reviewer for giving advice. We have reorganized the manuscript and added descriptions related to ovarian cancer to try to improve the quality of the article.

“ 3. Mechanism of MiRNA in Ovarian Cancer Invasion and Metastasis

Ovarian cancer is one of the most common gynecological malignancies, and more than 90% of ovarian cancers are epithelial cancers. Although treatments and strategies continue to improve, the 5-year survival rate for epithelial ovarian cancer is only 30%. In addition, ovarian cancer has no obvious symptoms in the early stage, and more than 60% of patients have abdominal or distant metastases when diagnosed with ovarian cancer [41, 42]. The process of miRNA and ovarian cancer invasion and metastasis has been highly valued, and the abnormal expression of miRNA is closely related to the invasion and metastasis of ovarian cancer.

3.1 Ovarian cancer invasion and metastasis

The invasion and metastasis of tumor cells can be divided into three independent stages: 1. Changes in tumor cell biochemistry, morphology and migration ability (changes in stem cell-like properties and migration patterns); 2. Surface receptors that mediate metastasis; The formation of tumor microenvironment around the target organ is conducive to the survival and proliferation of metastatic cells [43]. The tumor microenvironment plays an important role in the development of tumors. Its hypoxia, low pH and high pressure environment are beneficial to the growth factors, proteolytic enzymes and immunoinflammatory factors in the tumor cells. The process of metastasis, differentiation, etc. has an important influence [44], and the abnormal expression of miRNA can change the ability of tumor invasion and metastasis by regulating the extracellular matrix [45].

3.3. MiR-200 Family

The miR-200 family (miR-200a, miR-141, miR-200b, miR-200c, miR-429) is a newly discovered tumor metastasis-associated miRNA. It inhibits the expression of E-cadherin transcriptional inhibitors (zinc-finger enhancer binding 1) ZEB1 and ZEB2, increases the expression of E-cadherin, and promotes the transformation of mesenchymal cells into epithelial cells. The 3'UTR of ZEB1 and ZEB2 mRNA contains 3 and 2 miR-200a/141 potential binding sites, respectively. MiR-200a binds to the 3' UTR complementary series of ZEB1/ZEB2 mRNA via the 5' end, resulting in post-transcriptional inhibition of ZEB1/ZEB2 gene expression. In addition, downregulation of miR-200a expression in tumor cells increased ZEB1/ZEB2 expression levels and inhibited E-cadherin. The cells were induced to undergo EMT to promote invasion and metastasis of tumor cells; up-regulation of miR-200a expression blocked EMT and inhibited invasion and metastasis of tumor cells. However, ZEB1 and ZEB2 also inhibited the expression of the miR-200 family, partially reversing the mesenchymal–to-epithelial transition. Therefore, the ZEB gene family can be considered to form a dual negative regulatory pathway with the miR-200 family [55].

miR-429 belongs to one of the members of the miR-200 family, and the cells not only undergo morphological changes, but also transform from mesenchymal cells to epithelial cells after transfection of miR-429. EMT-related marker genes such as ZEB1, ZEB2, E-cadherin, N-cadherin, fibronectin 1 (FN1), and vimentin also change at mRNA levels [56]. Microarray detection also found that 296 genes were significantly down-regulated, such as the mesenchymal cell marker molecules ZEB1 and Versican (VCAN); while 373 genes were significantly up-regulated, including Tetraspanin 13 (TSPAN13), Caveolin 2 (CAV2), Desmoplakin (DSP), epithelium cell adhesion molecule (EPCAM). Therefore, miR-429 can be considered to play a key role in regulating EMT in ovarian cancer cells [57].

Thus, the miR-200 family differs in the expression of different tissues of ovarian cancer, at different stages, and in benign or malignant ovarian cancer. This suggests that it may induce ovarian cancer as a proto-oncogene, and involvement in ovarian cancer invasion and metastasis is closely related to poor prognosis. miR-200 has important molecular markers as a potential target for ovarian cancer invasion and metastasis and prognosis and treatment of ovarian cancer metastasis.

3.4. MiRNAs Regulate Expression and Function of Extracellular Matrix

The extracellular matrix constitutes a dynamic system of cellular mechanical changes and biochemical changes in extracellular fluid. The composition between cells and fibers includes proteoglycans, polyglycoproteins and glycosaminoglycans, and ECM molecules regulate cell migration and tumor metastasis[58]. Abnormal expression of miRNA can regulate the ability of tumor metastasis by degrading ECM [59]. The matrix metalloproteinases (MMPs) family is a class of zinc-dependent endopeptidases that degrade and remodel ECM to accelerate tumor metastasis [60]. MMPs are important proteases in tumor invasion. MMP-2, MMP-3, MMP-7 and MMP-9 are thought to be involved in the invasion and migration of tumor cells in ovarian cancer. In ovarian cancer patients, by comparing ovarian cancer tissues with distant non-ovarian cancer tissues, MMP-7 was significantly increased in ovarian cancer tissues, and miR-543 was significantly reduced. miR-543 inhibits MMP-7 transcription by binding to the 3'UTRs of MMP-7 mRNA and reduces metastasis of ovarian cancer cells [61]. In addition, miR-205 showed a dramatic increase in ovarian cancer in the late stage. Since miR-205 inhibits transcription factor-21 (TCF-21), it increases the ability of ovarian cancer cells to spread and metastasize [62]. In addition, the invasiveness of miR-205-transfected ovarian cancer cells was significantly increased 5-fold. Conversely, inhibition of miR-205 reduced cell invasiveness. This suggests that miR-205 plays a role in proto-oncogenes and promotes tumor invasion during ovarian cancer.

3.5. MiRNA in Angiogenesis

Tumor angiogenesis is an important process of tumor progression, which facilitates the dissemination and metastasis of tumor cells through blood circulation [43]. The miR-200 family is not only associated with EMT but also with tumor angiogenesis. It has been reported that the miR-200 family inhibits the formation of blood vessels by directly or indirectly targeting the interleukin-8 secreted by tumor epithelial cells and the chemokine CXCL-1. Transferring the miR-200 family to the tumor epithelium showed a significant decrease in tumor cell metastasis and angiogenesis, as well as normalization of blood vessels [63]. Vascular endothelial growth factor (VEGF) also plays an important role in promoting the invasion and metastasis of ovarian cancer. As a pro-angiogenic factor, it also promotes endothelial cell growth and migration while increasing vascular permeability. Moreover, VEGF also increases the expression of miR-205 in ovarian cancer cells, reduces Ezrin and lamin A/C, and promotes proliferation and metastasis of ovarian cancer cells[64].”

Round 2

Reviewer 2 Report

The authors have provided a revised manuscript with lots of helpful pieces of information regarding miRNA in cancer, with some specific emphasis on roles in ovarian cancer.

The authors have attempted to address many of the reviewers prior concerns and the effort is much appreciated. The article is struggling to focus on the chief purpose of a review, which in this reviewers eyes is to clearly synthesize,  consolidate and narrate the importance of the data, not just display the data. This is being lost to a large degree in this manuscript and this might be due to language barriers, so I am strongly recommending receiving grammatical and composition help to effectively achieve this objective. I have provided some recommendations and highlighted some specific issues to address in the manuscript, however the amount of change is extensive throughout the manuscript for this manuscript to be acceptable as a scholarly work. To be clear, the information provided is appropriate and important, it is the composition that needs work to achieve the goals of a review. 

I encourage the authors to spend significant time focusing on grammar and organization of the paper and resubmitting what will be a nice addition to the literature.

Several Grammatical Errors  in Added Text Ln 80-111

Some outlined below.

Ln 82- Revise grammar - The microprocessor is composed of ribonuclease Drosha and 81 microprocessor complex subunit 8 (DGCR8) constitutes a complex

Ln 87- 90 ku?

Ln91 - membrane to the cytoplasm for subsequent processing and processing

Ln.127 -145 - Organizationally I would put together all the overexpressed miRNA comments in one place and the all the underexpressed statements in another instead of bouncing back and forth. This will dramatically improve readability and comprehension.

Ln 147-152 says the same thing as Ln 133, these sections should be revised for conceptual and organizational clarity.

Section 2.3 and 2.4 - reads again like a laundry list without understanding how these genes that are targeted by miRNAs act as tumor suppressors - what is their biological function. It is good to have a list - maybe consider a more comprehensive table (more so than provided table 1) for this information to highlight your points.

Ln 202 - What process? I don't understand this sentence "The process of miRNA and ovarian cancer 202 invasion and metastasis has been highly valued,"

Are MiR-125A and EZRIN related, if not then what does miR-125A target? , same with miR-200 and mi-R429, 

Ln 216-Ln 229 Again maybe a single table for the whole paper that highlights overexpression, or underexpression, tumors where its observed, genes that are known to be targeted and process impacted in promoting or suppressing tumors (a more comprehensive and clear table than the Table 1 provided)  This could capture much of the information and allow the reader to extract the information the authors are struggling to convey in text

Ln 280 - In a review try to provide specifics and avoid ambigious statements. Instead of abnormal expression, states if its over or under expression, instead of saying "can" state under which conditions it does.  Otherwise  the statements lack impact and clarity, the goal of a review and instead is its all conceptual amorphous ambiguity. 

Ln 284 - 287 - This is an excellent example of good review of detail and mechanism.  This could be said in a more concise manner. Ln 276-283 are great intro then go straight into.  By comparing ovarian cancer tissues to health ovarian tissues it was observed that miR-543 which targets MMP7 was significantly reduced.

Ln 309 - 60-80%, try to clarify why the rates of advanced disease and outcomes are reported differently in different sections of the paper.  It maybe better to stick to a range from the beginning if multiple sources are being cited that have disparate observations, regardless try to make your text consistent throughout for clarity. 

Ln320 - again this information highlighted in a table or somehow connected to the previous conversation would help the reader integrate this information in a more meaningful way.

Ln - 329 is it known what miR30a-5p targets - how this might be related to cancer biology?

Ln 342-347 -Is a good example of context and direction to relay significance of alter miRNA expression. In contratst Ln 348 - 356 is conceptually confusing about the relationship and impact of miR-140-5P with PDGRA and cancer outcomes/progression. 

Ln 357-372  - incorporate these details into a table.  

Section 6.  Is interesting - shows the value of altered expression of miRNA in both directions - what do miR-200c and miR-141 target? Also this might be better up above because  biology associated with outcomes in cancer provides great rationale for study in the first place. Even if it is unknown which genes these miRNAs target, state that fact and use as rationale for  in depth review of miRNAs.  This observations makes a great introduction for inferences of miRNA in cancer biology, diagnosis and treatment in a short succinct paragraph. 

Author Response

Dear referee 2:

We would like to thank the Reviewers for his/her kind expression of our original work. In addition, all the authors are truly appreciative for the Reviewers’ effort, and thank the Reviewer s’ for his/her critical and highly constructive comments on how to improve this manuscript. We are grateful for the opportunity to revise our manuscript; we have endeavored to address the problems indicated by the Reviewer.

Comments and Suggestions for Authors

The authors have provided a revised manuscript with lots of helpful pieces of information regarding miRNA in cancer, with some specific emphasis on roles in ovarian cancer.

The authors have attempted to address many of the reviewers prior concerns and the effort is much appreciated. The article is struggling to focus on the chief purpose of a review, which in this reviewers eyes is to clearly synthesize, consolidate and narrate the importance of the data, not just display the data. This is being lost to a large degree in this manuscript and this might be due to language barriers, so I am strongly recommending receiving grammatical and composition help to effectively achieve this objective. I have provided some recommendations and highlighted some specific issues to address in the manuscript, however the amount of change is extensive throughout the manuscript for this manuscript to be acceptable as a scholarly work. To be clear, the information provided is appropriate and important, it is the composition that needs work to achieve the goals of a review.

I encourage the authors to spend significant time focusing on grammar and organization of the paper and resubmitting what will be a nice addition to the literature.

Several Grammatical Errors in Added Text Ln 80-111

Some outlined below.

Ln 82- Revise grammar - The microprocessor is composed of ribonuclease Drosha and 81 microprocessor complex subunit 8 (DGCR8) constitutes a complex

Response: We thank the reviewer for checking our article in detail and indicating the unclear information. We have corrected the mistake. We use red letters to indicate where the changes have been made.

Ln 87- 90 ku?

Response: We thank the reviewer for pointing out the inappropriate description. We have corrected the mistake.

Ln91 - membrane to the cytoplasm for subsequent processing and processing

Response: We thank the reviewer for pointing out the inappropriate description. We have corrected the mistake.

Ln.127 -145 - Organizationally I would put together all the overexpressed miRNA comments in one place and the all the underexpressed statements in another instead of bouncing back and forth. This will dramatically improve readability and comprehension.

Response: We thank the reviewer for giving advice. We have added the latest regulatory mechanisms of miRNA in the invasion and metastasis of ovarian cancer.

Ln 147-152 says the same thing as Ln 133, these sections should be revised for conceptual and organizational clarity.

Response: We thank the reviewer for pointing out the inappropriate description. We have corrected the mistake.

Section 2.3 and 2.4 - reads again like a laundry list without understanding how these genes that are targeted by miRNAs act as tumor suppressors - what is their biological function. It is good to have a list - maybe consider a more comprehensive table (more so than provided table 1) for this information to highlight your points.

Response: We thank the reviewer for giving advice. We have added the latest regulatory mechanisms of miRNA in the invasion and metastasis of ovarian cancer. We have merged the data into Table 1.

Ln 202 - What process? I don't understand this sentence "The process of miRNA and ovarian cancer 202 invasion and metastasis has been highly valued,"

Response: We thank the reviewer for pointing out the inappropriate description. We have corrected the mistake.

Are MiR-125A and EZRIN related, if not then what does miR-125A target? , same with miR-200 and mi-R429,

Response: All authors thank the Reviewer for indicating the unclear point. There is currently no direct indication that miR125 regulates miR200 and miR429. But all of the above are related to cancer metastasis and EMT.

Ln 216-Ln 229 Again maybe a single table for the whole paper that highlights overexpression, or underexpression, tumors where its observed, genes that are known to be targeted and process impacted in promoting or suppressing tumors (a more comprehensive and clear table than the Table 1 provided)  This could capture much of the information and allow the reader to extract the information the authors are struggling to convey in text

Response: We thank the reviewer for giving advice. We have added the latest regulatory mechanisms of miRNA in the invasion and metastasis of ovarian cancer. We have merged the data into Table 1.

Ln 280 - In a review try to provide specifics and avoid ambigious statements. Instead of abnormal expression, states if its over or under expression, instead of saying "can" state under which conditions it does. Otherwise the statements lack impact and clarity, the goal of a review and instead is its all conceptual amorphous ambiguity.

Response: We thank the reviewer for pointing out the inappropriate description. We have corrected the mistake.

Ln 284 - 287 - This is an excellent example of good review of detail and mechanism.  This could be said in a more concise manner. Ln 276-283 are great intro then go straight into.  By comparing ovarian cancer tissues to health ovarian tissues it was observed that miR-543 which targets MMP7 was significantly reduced.

Response: We thank the review committee for their appreciation.

Ln 309 - 60-80%, try to clarify why the rates of advanced disease and outcomes are reported differently in different sections of the paper. It maybe better to stick to a range from the beginning if multiple sources are being cited that have disparate observations, regardless try to make your text consistent throughout for clarity.

Response: We thank the reviewer for checking our article in detail. We have corrected the mistake.

Ln320 - again this information highlighted in a table or somehow connected to the previous conversation would help the reader integrate this information in a more meaningful way.

Response: We thank the reviewer for giving advice. We have merged the data into Table 1.

Ln - 329 is it known what miR30a-5p targets - how this might be related to cancer biology?

Response: We thank the reviewer for indicating the unclear information. miR30a-5p is currently only clearly labeled in colorectal cancer and prostate cancer, and there are no reports of ovarian cancer. (Cancer Biomark 2019. doi: 10.3233/CBM-182129: Artif Cells Nanomed Biotechnol. 2019. 47(1):278-289. doi: 10.1080/21691401.2018.1553783.)

Ln 342-347 -Is a good example of context and direction to relay significance of alter miRNA expression. In contrast Ln 348 - 356 is conceptually confusing about the relationship and impact of miR-140-5P with PDGRA and cancer outcomes/progression.

Response: We thank the reviewer for indicating the unclear information. This section shows that miR-140-5P has the function of negatively regulating PDGFRA. However, the detailed regulation mechanism still needs to be clarified, so it is impossible to conclude that miR-140-5P affects the development of cancer by PDGFRA.

Ln 357-372 - incorporate these details into a table. 

Response: We thank the reviewer for giving advice. We have merged the data into Table 1.

Section 6.  Is interesting - shows the value of altered expression of miRNA in both directions - what do miR-200c and miR-141 target? Also this might be better up above because biology associated with outcomes in cancer provides great rationale for study in the first place. Even if it is unknown which genes these miRNAs target, state that fact and use as rationale for in depth review of miRNAs. This observation makes a great introduction for inferences of miRNA in cancer biology, diagnosis and treatment in a short succinct paragraph.

Response: We thank the reviewer for indicating the unclear information. Currently, the two genes may be associated with EMT, but no clear regulatory mechanisms have been reported in ovarian cancer. (Theranostics. 2018. 8(21): 5801-5813. Cell Death Dis. 2019.10(2): 129.)

The authors have a contributed a thoughtful manuscript outlining the role of miRNA's in the diagnosis, contribution and amelioration of ovarian cancer. The manuscript is written with a sense of technical proficiency but the text wanders with keeping in intention (ovarian cancer) set out in title and abstract.  Also the role of miRNAs mostly reads like a grocery/laundry list of differentiatial expression and could use some thoughtful synthesis of the actually biological pathways regulated by RNAs. Specific examples are given below. With additional work this could be an excellent resource for others. 

Response: We thank the reviewer for giving advice. We have reorganized the manuscript and added descriptions related to ovarian cancer to try to improve the quality of the article.